# Phytochemical Profiles and Antioxidant Activity of Grasses Used in South African Traditional Medicine

**DOI:** 10.3390/plants9030371

**Published:** 2020-03-17

**Authors:** Fikisiwe Gebashe, Adeyemi O. Aremu, Jiri Gruz, Jeffrey F. Finnie, Johannes Van Staden

**Affiliations:** 1Research Centre for Plant Growth and Development, School of Life Sciences, University of KwaZulu-Natal Pietermaritzburg, Private Bag X01, Scottsville 3209, South Africa; fix.gebs@gmail.com (F.G.); Oladapo.Aremu@nwu.ac.za (A.O.A.); finnie@ukzn.ac.za (J.F.F.); 2Indigenous Knowledge Systems (IKS) Centre, Faculty of Natural and Agricultural Sciences, North West University, Private Bag X2046, Mmabatho 2735, South Africa; 3Laboratory of Growth Regulators, The Czech Academy of Sciences, Institute of Experimental Botany & Palacký University, Šlechtitelů 27, 78371 Olomouc, Czech Republic; jiri.gruz@upol.cz

**Keywords:** flavonoids, medicinal plants, phenolic acids, Poaceae, secondary metabolites, UHPLC

## Abstract

Grasses are a valuable group of monocotyledonous plants, used as nourishing foods and as remedies against diseases for both humans and livestock. Phytochemical profiles of 13 medicinal grasses were quantified, using spectrophotometric methods and ultra-high-performance liquid chromatography–tandem mass spectrometry (UHPLC–MS), while the antioxidant activity was done using DPPH and ferric-reducing-power assays. The phytochemical analysis included the total soluble phenolic content, flavonoids, proanthocyanidins, iridoids and phenolic acids. Among the 13 grasses, the root methanolic extracts of *Cymbopogon* spp., *Cymbopogon nardus* and *Cenchrus ciliaris* contained the highest concentrations of total soluble phenolics (27–31 mg GAE/g DW) and flavonoids (4–13 mg CE/g DW). Condensed tannins and total iridoid content were highest (2.3 mg CCE/g DW and 3.2 mg HE/g DW, respectively) in *Cymbopogon nardus*. The most common phenolic compounds in the grass species included *ρ*-coumaric, ferulic, salicylic and vanillic acids. In the DPPH radical scavenging assay, the EC_50_ values ranged from 0.02 to 0.11 mg/mL for the different grasses. The best EC_50_ activity (lowest) was exhibited by *Cymbopogon nardus* roots (0.02 mg/mL) and inflorescences (0.04 mg/mL), *Cymbopogon* spp. roots (0.04 mg/mL) and *Vetiveria zizanioides* leaves (0.06 mg/mL). The highest ferric-reducing power was detected in the whole plant extract of *Cynodon dactylon* (0.085 ± 0.45; r^2^ = 0.898). The observed antioxidant activity in the various parts of the grasses may be due to their rich pool of phytochemicals. Thus, some of these grasses provide a source of natural antioxidants and phytochemicals that can be explored for their therapeutic purposes.

## 1. Introduction

The value of medicinal plants lies in their chemical constituents, which often possess and/or influence biological activities [1]. Generally, compounds from plants are classified into primary and secondary metabolites. The role of secondary metabolites in plants include protection against insects, herbivores and pathogens (biotic) and/or to ensure survival under abiotic stresses [2,3,4]. They are also attractants for pollinators and seed-dispersing animals, allelopathic agents, UV protectants and signal molecules in the formation of nitrogen-fixing root nodules in legumes [5]. These chemical defense substances have received considerable attention due to their potential as pharmaceuticals, food additives and valuable chemicals for human use [2].

Members of the Poaceae family have been characterized as being dominated by alkaloid-poor species and have few medicinal uses [6,7]. Moreover, this family of plants produces limited chemical defenses to avoid herbivory and mostly rely on their physical defenses, such as silicates and leaf toughness [8]. Therefore, few species in the family Poaceae produce chemical compounds that are considered to be of medicinal value. Grasses contain the types of secondary metabolites that are regarded as biologically inactive and/or have low activity (e.g., tannins and lignin). These compounds are often characterized by high molecular weight; thus, they reduce digestibility and, as a result, herbivory [8]. Regardless of these aforementioned concerns, there are still many grasses that are often utilized in South African traditional medicine.

When compared to other plant families, few studies have evaluated the pharmacological and phytochemical properties of grasses and/or compounds isolated from grass extracts [8]. Nevertheless, grasses possess antimicrobial, antimutagenic, antitumor, antidiarrheal, antidiabetic, anti-inflammatory, anti-plasmodial, diuretic, hepato-protective and antioxidant activities [9]. More than 100 secondary metabolites have been identified and isolated in different grass species used in South African traditional medicine [9]. These compounds include steroids, phenol aldehydes (benzaldehyde derivatives and cinnamaldehyde derivatives), benzoxazinoid derivatives, chiral monoterpenes, aldehydes, fatty acids and volatiles; and they have exhibited anticancer, antimicrobial, anti-inflammatory and antioxidant potential [10,11,12,13]. The compounds in grasses have been shown to demonstrate certain degrees of biological activity, thereby contributing to their therapeutic claims in traditional medicine. Thus, the study was aimed at evaluating preliminary phytochemical potential (such as total soluble phenolics, flavonoids, iridoids and condensed tannins) and antioxidant activity (DPPH radical-scavenging activity and ferric-reducing power) of 13 grasses that were documented during a previous ethnobotanical survey as remedies in South African traditional medicine [14]. The methods used for preliminary evaluation of phytochemicals and pharmacological activities were aimed at providing the basic evidence that may explain the rationale for the use of these grasses in traditional medicine. There is very little scientific (pharmacological and phytochemical properties) evidence available for their use in traditional medicine.

## 2. Results and Discussion

### 2.1. Phytochemical Evaluation

In this study, total soluble phenolic content of South African medicinal grasses ranged from 4.2 to 30.9 mg GAE/g DW (Figure 1). The highest total soluble phenolic content was obtained in methanolic root extracts of *Cymbopogon* spp., *Cymbopogon nardus* (L.) Rendle and *Cenchrus ciliaris* (L.). Similarly, the highest flavonoid contents (Figure 2) were obtained in the same species. However, the highest concentrations of iridoid content and condensed tannins (Figure 3 and Figure 4) were observed in *Cymbopogon* species. Several studies have reported the beneficial effects, biological activities and/or preventative effects of phenolics [10], flavonoids [11] and iridoids [12,13]; against the development of chronic degenerative diseases, and as a treatment for most of the free-radical-related disorders [14]. *Cymbopogon* species are distinguished by having the highest content for most of the evaluated phytochemicals, which may strongly influence and contribute to their resultant biological activities.

Some of the flavonoids isolated from *Cymbopogon* species include isoorientin and tricin from the whole plant of *Cymbopogon parkeri* Stapf. [15]; luteolin, luteolin 7-O-glucoside (cynaroside), isoscoparin and 2′-O-rhamnosyl isoorientin from the leaves and rhizomes of *Cymbopogon citrates* (DC. ex Nees) Stapf. Other flavonoid compounds isolated from the aerial parts of *Cymbopogon citratus* are quercetin, kaempferol and apigenin, elemicin, catechol, chlorogenic acid, caffeic acid and hydroquinone [16]. These compounds may also be found in the reported species due to their chemotaxonomic relationship. As indicated by Chesselet et al. [17], a number of studies report on the occurrence of tannin-like substances in the epidermal cells of southern African tropical grasses. Furthermore, the authors stated that condensed tannins have been confirmed in grasses, using chemical analysis. Though relatively low quantities of condensed tannins were observed in this study, *Cymbopogon* species had the highest amounts. Tannins are a group of phenolic metabolites in many woody and some herbaceous higher plant species. They have been investigated for their wide range of pharmacological and phytochemical properties, which contribute to their efficacy in traditional medicine [18]. Other evaluated phytochemicals were iridoid compounds, which are a large group of monoterpenoids with a cyclopenta [c] pyranoid skeleton, and are often intermediates in the biosynthesis of alkaloids [12,19]. These compounds have been associated with pharmacological activities, such as antimicrobial, analgesic, antioxidant, sedative and anti-inflammatory activities [12].

*Sporobolus africanus* (Poir.) Robyns and Tournay, *Sporobolus pyramidalis* (Beauv.) and *Eragrostis curvula* (Schrad.) Nees root extracts exhibited the lowest concentration of phenolics. Despite the low levels of phytochemical compounds in some grasses, overall the grass species showed good antioxidant activity [10]. Hence, the results reveal that different plant species possess diverse biological compounds (which result in various biological activities), in differing quantities. Their efficacy or mode of action varies depending on the type of compounds they contain and the pathogens they act against, e.g., *Coix lacryma-jobi* L., *Cynodon dactylon* (L.) Pers., and *Setaria megaphylla* (Steud.) T. Durand and Schinz had low concentrations of phenolics, flavonoids, iridoid content and condensed tannins (Figure 1, Figure 2, Figure 3 and Figure 4) but yielded good antibacterial activity [20]. However, limited work has been done on the phytochemical evaluation of grasses, thus allowing minimal comparative data to the current findings.

#### Concentration of Phenolic Acids

Grass extracts exhibited diverse categories and quantities of phenolic compounds in different plant parts (Table 1 and Table 2). Soluble phenolic compounds quantified included gallic, chlorogenic, *p*-coumaric, protocatechuic, caffeic, sinapic, vanillic, ferulic, *p*-hydroxybenzoic, syringic and salicylic acids. The highest concentration of phenolic acids occurred in the leaf extracts (Table 1 and Table 2) of *Setaria megaphylla*, *Panicum maximum*, *Cenchrus ciliaris*, *Coix lacryma*-*jobi* and *Eragrostis curvula*. Apart from *Setaria megaphylla* root (dichloromethane, DCM) extract, which showed noteworthy antibacterial activity, these grass species exhibited weak antibacterial activity [20]. In addition, these species had weak antioxidant activity, regardless of the quantity of their phytochemicals.

Root extracts of *Coix lacryma*-*jobi*, *Setaria megaphylla*, *Cenchrus ciliaris*, *Cymbopogon nardus*, *Cymbopogon* spp., *Panicum maximum* and *Eragrostis curvula* also had high quantities of phenolic acids. Gallic acid was the least represented compound among the evaluated phenolic acids, in whole plant samples of *Cynodon dactylon*; inflorescence of *Eragrostis curvula*; in roots and leaves of *Cymbopogon nardus*, *Sporobolus africanus* and *Vetiveria zizanioides*; and in roots of *Eragrostis curvula*, *Panicum maximum* and *Setaria megaphylla*. However, across the evaluated grass species, the most common phenolic compounds were *p*-coumaric, ferulic, salicylic and vanillic acids. Hartley and Morrison [21] also evaluated the monomeric and dimeric phenolic acids released from cell walls of the tall fescue *Festuca arundinacea* and coastal bermudagrass *Cynodon dactylon*. Similar to the results obtained in this study, *p*-coumaric and ferulic acids were the major monomers extracted. Kroon and Williamson [22] also reviewed the importance of hydroxycinnamates (caffeic, *p*-coumaric and sinapic acids) as antioxidant agents and their additional health benefits in the diet. Ferulic acid has been reported to have many pharmacological properties, including antioxidant [23,24], antimicrobial, anti-inflammatory, antithrombosis and anticancer activities [25]. Rice-Evans et al. [26] reviewed the importance of phenolic acids, along with their biological activities, relative to their structural features. The results showed that antioxidant activity of monophenols is increased by one or two methoxy substitutions that occur in the *ortho* or *para* to the OH position in phenol or phenolic acid (ferulic acid). For instance, sinapic acid has better activity than ferulic acid, which has better activity than *p*-coumaric acids. Relative to the current findings, *Cymbopogon* species had high quantities of both *p*-coumaric and ferulic acids, which might have resulted in the higher DPPH scavenging activity (Table 3). Cinnamic acids have also been reported to be more active than the carboxylate group in the benzoic acids, because of their ability to donate H^+^. Therefore, hydroxycinnamic acids (caffeic, sinapic, ferulic and *p*-coumaric acids) are more active than hydroxybenzoic acids (protocatechuic, syringic, vanillic and *p*-hydroxybenzoic) [27,28]. Phenolic acids of the evaluated grass species showed positive correlation with their antioxidant activity. Thus, supporting the obtained antioxidant activity of the *Cymbopogon* species with their phenolic acids

### 2.2. Antioxidant Activity

The DPPH radical-scavenging activity and the ferric-reducing-power assays were used to determine the antioxidant activity of grass extracts. These are two different methods which measured the free radical scavenging potential (DPPH) and the ability to reduce iron (ferric-reducing power) of extracts [29,30], which can be used to combat oxidative stress responsible for numerous diseases.

#### 2.2.1. DPPH Radical-Scavenging Activity

Phenolic compounds such as flavonoids, phenolic acids and tannins have diverse biological properties, especially antioxidant activity. The absence of antioxidants responsible to quench reactive radicals has been observed to facilitate the development of degenerative diseases [31], cancers [32], neurodegenerative diseases, Alzheimer’s [33] and inflammatory diseases [34]. Therefore, the importance of natural antioxidants cannot be overemphasized, as they serve as preventative medicines.

There was an increase in the free radical scavenging activity with an increase in extract concentration (results not shown). The EC_50_ values showed the best activity (lowest) in *Cymbopogon nardus* roots and inflorescence, *Cymbopogon* spp. roots (0.02, 0.04 and 0.04 mg/mL) and in leaves of *Cymbopogon nardus* and *Vetiveria zizanioides* (0.06 mg/mL). The lowest scavenging activity (EC_50_ 0.11 mg/mL) was observed in roots of *Eragrostis curvula*, *Panicum maximum*, *Setaria megaphylla* and *Vetiveria zizanioides*; leaves of *Sporobolus africanus* and *Coix lacryma-jobi*; and whole plant of *Cynodon dactylon* (EC_50_ 0.10 mg/mL), when compared to the other grass species. Overall, the evaluated grass species had good antioxidant activity, ranging from 0.02 to 0.11 mg/mL. The activity of the plant extracts was comparable with the reference antioxidant (ascorbic acid) with EC_50_ of 0.025 mg/mL. Thus, it can be deduced that the observed antioxidant activity exhibited by leaves, roots and inflorescence of the different species, in particular the *Cymbopogon* species (Table 3), was due to their high phytochemical compounds [35,36]. Wojdyło et al. [37] evaluated several herbs and showed that species which were rich in phenolic compounds had noteworthy antioxidant activity. They evaluated the free radical scavenging activity of *Vetiveria zizanioides* oil, which exhibited an EC_50_ of 7.79 mg/mL, a result which was higher than that obtained in this study (Table 3). The difference in antioxidant activity may be due to different types of extracts used, which were leaf extracts and the oil of *Vetiveria zizanioides*.

#### 2.2.2. Iron-Reducing Power

There was an increase in absorbance values with an increase in concentration (Table 4). *Sporobolus africanus* (leaves and roots) and *Sporobolus pyramidalis* root extracts showed a strong reducing power (slope = 0.086–0.124), comparable to that of a standard antioxidant (ascorbic acid). Though the results obtained in the DPPH assay were different from the FRAP analysis, most grass species showed noteworthy antioxidant activity. *Eragrostis curvula* leaves, *Coix lacryma-jobi* roots and *Setaria megaphylla* leaves exhibited weak reducing power, with a slope of 0.051, 0.054 and 0.059, respectively. The possible explanation for the variations in the results yielded by the two methods may be linked to the type and quantities of phytochemical compounds in the grass extracts. Some authors linked these differences with the mechanism involved in antioxidant reactions and also principles of the different assays and experimental conditions [38,39]. However, in relation to this study, DPPH and FRAP are both categorized in single electron-transfer reactions [40]. Though Prior et al. [41] classified DPPH into a category that uses both single-electron transfer (SET) and hydrogen transfer (HAT) reactions, which might account for some differences between the two methods.

## 3. Materials and Methods

### 3.1. Chemicals

Standards of phytochemicals (gallic acid, catechin, harpagoside, cyanidin chloride and ascorbic acid), phenolic acids (gallic acid, 3,5-dihydroxybenzoic acid, protocatechuic acid, chlorogenic acid, gentisic acid, 4-hydroxybenzoic acid, caffeic acid, vanillic acid, syringic acid, 3-hydroxybenzoic acid, 4-courmaric acid, sinapic acid, ferulic acid, 3-courmaric acid, 2-courmaric acid, salicylic acid and trans-cinnamic acid), 1,1-diphenyl-2-picrylhydrazyl, vanillin, aluminum chloride, sodium hydroxide, ferric ammonium sulphate and leucocyanidin were obtained from Sigma-Aldrich Fine Chemicals (St. Louis, MO, USA). Deuterium-labeled standards of 4-hydroxybenzoic acid (2,3,5,6-D4) and salicylic acid (3,4,5,6-D4) were obtained from Cambridge Isotope Laboratories (Andover, MA, USA). Folin–Ciocalteu’s phenol reagent, sodium carbonate, sulphuric acid, butanol, hydrochloric acid, formic acid and methanol were purchased from MERCK (Darmstadt, Germany).

### 3.2. Plant Material and Extract Preparation

Grasses were collected from three locations, namely the University of KwaZulu-Natal, Pietermaritzburg Campus (29.6196° S, 30.3960° E, grasslands and Ukulinga farm), Durban (29.8587° S, 31.0218° E) and Zululand (27.8872° S, 31.4456° E) in KwaZulu-Natal Province, South Africa. Voucher specimens were deposited in the Bews Herbarium at the University of KwaZulu-Natal, Pietermaritzburg. For the morphological appearance of some of the collected grasses, see Appendix A. Grasses were separated into inflorescences, leaves, and roots and/or used as a whole plant. Plants were oven-dried at 50 °C for 3 days, ground into fine powders and stored in airtight containers, at room temperature, in the dark. Three replicates of finely ground plant samples (0.1 g) were extracted with 10 mL of 80% methanol on ice in a sonication bath for 1 h. The methanolic extracts were filtered under vacuum, through Whatman No. 1 filter paper, and the filtrate was immediately used for the determination of total soluble phenolics, flavonoids, iridoid content and condensed tannins (proanthocyanidins).

For the Ultra-High-Performance Liquid Chromatography (UHPLC), ground material was homogenized with 80% methanol, using an oscillation ball mill (MM 301, Retsch, Haan, Germany), at a frequency of 27 Hz, for 3 min. The extracts were then centrifuged at 20,000 rpm for 10 min, and the supernatant used for UHPLC.

Grass extracts used to determine antioxidant activity were obtained from 2 g ground material extracted in 20 mL of 80% methanol, by sonication (Branson Model 5210, Branson Ultrasonics B.V., Soest, The Netherlands) in ice-cold water, for 1 h. The extracts were then filtered through Whatman No. 1 filter paper and concentrated in vacuo, using a rotary evaporator (Büchi, Germany) at 30 °C. The concentrates were transferred to pre-weighed glass vials and completely dried under a stream of cold air, at room temperature. Once a constant weight of each extract was obtained, the extracts were stored in the dark, at 10 °C, until used for analysis.

### 3.3. Phytochemical Evaluation

#### 3.3.1. Quantification of Total Soluble Phenolic Content

Total soluble phenolic in the grass extracts of 80% methanol was quantified, using the Folin and Ciocalteu (Folin-C) assay, as described by Makkar [42], with modifications. Gallic acid was used to generate the standard curve, and the assay was done in triplicate. Gallic acid was used as the standard for plotting the calibration curve. Total soluble phenolic content was expressed in mg gallic acid equivalents (GAE) per g dry weight (DW).

#### 3.3.2. Quantification of Flavonoid Content

Flavonoid content was determined, using the spectrophotometric assay, based on aluminum chloride complex formation, as described by Zhishen et al. [43], with modifications. The concentration of flavonoids in the test samples was measured in triplicate. Catechin was used as a standard for the calibration curve, and total flavonoid content was expressed in mg catechin equivalents (CE) per g DW.

#### 3.3.3. Quantification of Iridoid Content

Iridoid content was quantified by using a colorimetric method as described by Levieille and Wilson [44], which was adapted from Haag-Berrurier et al. [45]. The method was based on the characteristics of glucoiridoids to form a fulvoiridoid complex when reacted with aldehydes (such as vanillin) in an acidic medium [44]. The resultant complex gives a distinct dark-pink color during a spontaneous reaction at room temperature. HPLC-grade harpagoside was used as a standard for the calibration curve. Extracts were quantified in triplicate. The reaction was done at room temperature, and absorbance values were measured at 538 nm, using a UV–visible spectrophotometer (Varian, Australia). Total iridoid content was expressed as mg harpagoside equivalents (HE) per gram dry weight (DW).

#### 3.3.4. Quantification of Condensed Tannins

Condensed tannins (proanthocyanidins) were determined by using the butanol–HCl assay as described by Makkar [42], with modifications. A standard curve of cyanidin chloride was used to determine the concentration of condensed tannins per gram of each plant material. The quantity of condensed tannins was expressed as mg of cyanidin chloride equivalents (CCE) per gram dry weight (g DW) of plant material.

### 3.4. Ultra-High-Performance Liquid Chromatography–Tandem Mass Spectrometry (UHPLC–MS/MS)-Based Phenolic Acid Analysis

#### Instrumentation and Conditions

Samples were analyzed, using a UHPLC (Waters, Milford, MA, USA) apparatus coupled with a PDA 2996 photo diode array detector (PDA, Waters, Milford, MA, USA) and a Micromass Quattro *micro*^TM^ API benchtop triple quadrupole mass spectrometer (Waters, MS Technologies, Manchester, UK), equipped with a Z-spray electrospray ionization (ESI) source operating in negative mode. Instrumentation control, data collection and processing were completed, using MassLynx^TM^ software (version 4.0, Waters, Milford, MA, USA).

Chromatographic conditions of UHPLC and MS/MS settings were as described by Gruz et al. [46]. Phenolic acids of grasses were evaluated as detailed by Gruz et al. [46]. The multiple-reaction-monitoring mode was used for quantification, and analysis was done in triplicate, through technical replication. Grass extracts (supernatants, 20 mg/1.5 mL) were filtered through 0.45 μm nylon membrane filters (Alltech, Breda, The Netherlands) and injected into a reversed-phase column (BEH C_8_, 1.7 μm, 2.1 × 150 mm, Waters, Milford, MA, USA) and incubated at 30 °C. The sequence of linear gradients and isocratic flows of the mobile phase had a sequencing of 9.5 min, with solvent B (acetonitrile) balanced with solvent A (aqueous 7.5 mM HCOOH), at a flow rate of 250 µL/min (Table 5). After sequencing, the column was subsequently equilibrated under initial conditions for 2.5 min, under pressure ranging from 4000 to 8000 psi during the chromatographic run. The eluent was inserted into a PDA detector (scanning range 210–600 nm, resolution 1.2 nm) and then through an electrospray source exhibiting a source block temperature of 100 °C, desolvation temperature of 350 °C, capillary voltage of 2.5 kV and cone voltage of 25 V. Argon was applied as the collision gas (collision energy 16 eV), and nitrogen as a desolvation gas (500 L/h). Different retention windows were used for quantification, and analysis was done in triplicate through technical replication.

### 3.5. Determination of Antioxidant Activity

#### 3.5.1. 1-Diphenyl-2-picrylhydrazyl (DPPH) Radical-Scavenging Activity

The antioxidant activity of the grass extracts was evaluated using the DPPH assay as described by Karioti et al. [47] with modifications. Three concentrations (10, 25 and 50 mg/mL) of each dry extract were prepared by resuspending in 80% methanol for a working stock. In triplicate, 15 µL of methanolic extracts at different concentrations were diluted with 80% methanol, to a final volume of 750 μL. The diluted extracts were then added to an equal volume of DPPH (100 μM in methanol). The mixtures were incubated at room temperature, in the dark, for 30 min. A solution consisting of methanol in place of the extract was used as a negative control, while ascorbic acid (AsA) was used as a positive control. The absorbance was read at 517 nm, using a UV-visible spectrophotometer. Background correction of the extract absorbance was done by adding methanol in place of the DPPH solution; this was done for each extract in order to correct any absorbance due to extract color. The radical scavenging activity was calculated by using the following equation:(1)$$\%\;RSA=\left[1-\left(\frac{Aextract-Abackground}{Acontrol}\right)\right]*100$$
where *Aextract* is the absorbance of the reaction mixture containing the sample extract or standard antioxidant, *Abackground* is the absorbance of the background solution and *Acontrol* is the negative control. The EC_50_, which is the concentration of the extract required to scavenge 50% of DPPH radical, was determined for each extract, using GraphPad Prism software.

#### 3.5.2. Ferric-Reducing-Power Assay

The ferric-reducing power (FRAP) of the grass extracts was determined as described by Lim et al. [48], with modifications. Dried extracts and the positive controls (ascorbic acid, AsA and butylated hydroxytoluene, BHT) were re-dissolved in 80% aqueous methanol, to a concentration of 50 mg/mL. Thereafter, 30 µL of each plant extract, ascorbic acid or BHT was added to a 96-well micro-plate and serially diluted. Subsequently, potassium phosphate buffer (40 µL, 0.2 M, pH 7.2) and potassium ferricyanide (40 µL, 1% *w*/*v*) were added. The reaction mixtures were incubated at 50 °C for 20 min. After the incubation period, trichloroacetic acid (40 µL, 10% *w*/*v*), distilled water (150 µL) and FeCl_3_ (30 µL, 0.1% *w*/*v*) were added, followed by a second incubation, at room temperature, for 30 min, in the dark. Absorbance was measured at 630 nm using a micro-plate reader (Opsys MR™ micro-plate reader, Dynex Technologies Inc., Chantilly, VA, USA). The ferric-reducing-power capacities of the plant extracts and standard antioxidants were expressed graphically by plotting absorbance against concentration. Samples for the assay were prepared in triplicate. The absorbances of all samples were plotted against their concentrations, and the slope values for the samples were also determined.

### 3.6. Data Analysis

Data were subjected to analysis of variance (ANOVA), using SPSS for Windows (SPSS^®^, version 23.0. Armonk, New York, NY, USA). For the level of statistical significance (*p* ≤ 0.05), the mean values were further separated by using Duncan’s Multiple Range Test (DMRT).

## 4. Conclusions

Grass species, in particular *Cymbopogon* spp., have a rich pool of phytochemicals. They exhibit high concentrations of total soluble phenolics, flavonoids, iridoids and proanthocyanidins. UHPLC–MS/MS yielded eleven phenolic acids in the evaluated grass species. These included gallic, chlorogenic, *p*-coumaric, protocatechuic, caffeic, sinapic, vanillic, ferulic, *p*-hydroxybenzoic, syringic and salicylic acids. The main phenolic acids identified were *p*-coumaric, ferulic and chlorogenic acids in the roots of the evaluated grass species. Even though other phenolic acids occurred in low concentration, ranging from 0.1 to 19 μg/g DW, gallic acid was found in the lowest quantities in all the plant parts. Antioxidant activity in the DPPH assay was high in the *Cymbopogon* species; these grass species also had high amounts of total soluble phenolic content (Figure 1) and phenolic acids. Roots and inflorescence of *Cymbopogon nardus* and roots of *Cymbopogon* spp. had good free radical scavenging activity. The observed antioxidant activity in the various parts of the grass plants may be a result of their phytochemicals. Nevertheless, the evaluation of phenolic acids and their biological activities is imperative, as they possess extensive industrial application in pharmaceuticals, food and cosmetics. Exceptional biological activities in in vitro tests does not confirm that active plants can be used in traditional medicine but does provide basic evidence of their efficacy. Therefore, medicinal plants need to be further tested to determine their biological efficacies and safety. Future studies on the toxicology of active extracts, as well as isolation of active compounds from the extracts of plants, such as *Cymbopogon* species, are imperative.

## Figures and Tables

**Figure 1 plants-09-00371-f001:**
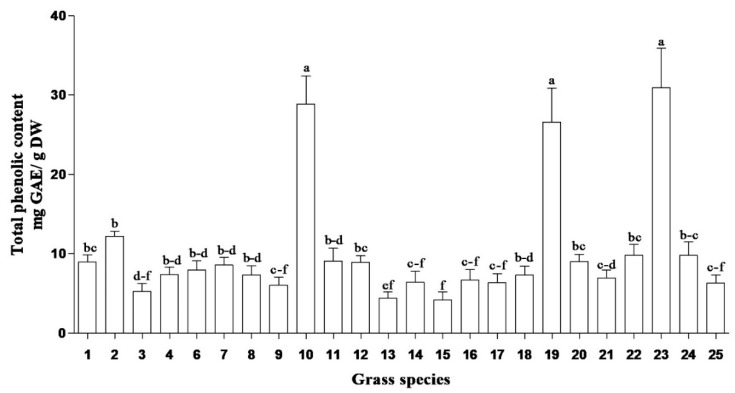
Total soluble phenolic content (mg GAE/g DW) in grass species used in South African traditional medicine. GAE = gallic acid equivalents; DW = dry weight. Bar represents mean ± standard error, *n* = 3. Bars with different letter(s) are significantly (*p* ≤ 0.05) different based on Duncan’s multiple range test (DMRT). (1) *Coix lacryma-jobi* Roots, (2) *Coix lacryma-jobi* Leaves, (3) *Eragrostis curvula* Roots, (4) *Eragrostis curvula* Leaves, (6) *Setaria megaphylla* Roots, (7) *Setaria megaphylla* Leaves, (8) *Vetiveria zizanioides* Roots, (9) *Vetiveria zizanioides* Leaves, (10) *Cymbopogon nardus* Roots, (11) *Cymbopogon nardus* Leaves, (12) *Cymbopogon nardus* Inflorescence, (13) *Sporobolus pyramidalis* Roots, (14) *Sporobolus pyramidalis* Leaves, (15) *Sporobolus africanus* Roots, (16) *Sporobolus africanus* Leaves, (17) *Imperata cylindrical* Roots, (18) *Imperata cylindrical* Leaves, (19) *Cenchrus ciliaris* Roots, (20) *Cenchrus ciliaris* Leaves, (21) *Panicum maximum* Roots, (22) *Panicum maximum* Leaves, (23) *Cymbopogon* spp. Roots, (24) *Cymbopogon* spp. Leaves and (25) *Cynodon dactylon* whole plant.

**Figure 2 plants-09-00371-f002:**
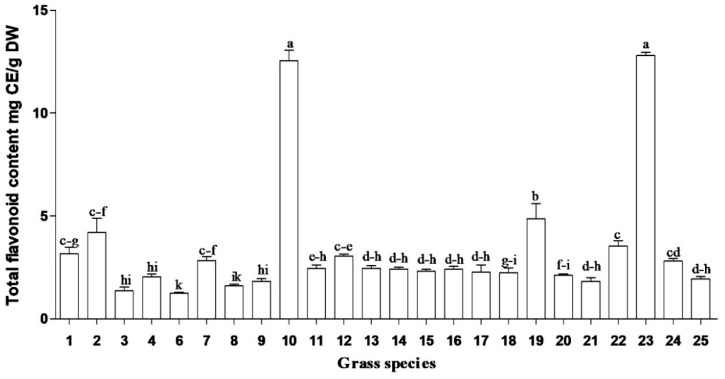
Concentrations of flavonoids in grass species used in traditional South African medicine. CE = Catechin equivalents; DW = dry weight. Bar represents mean ± standard error, *n* = 3. Bars with different letter(s) are significantly (*p* ≤ 0.05) different based on Duncan’s multiple range test (DMRT). (1) *Coix lacryma-jobi* Roots, (2) *Coix lacryma-jobi* Leaves, (3) *Eragrostis curvula* Roots, (4) *Eragrostis curvula* Leaves, (6) *Setaria megaphylla* Roots, (7) *Setaria megaphylla* Leaves, (8) *Vetiveria zizanioides* Roots, (9) *Vetiveria zizanioides* Leaves, (10) *Cymbopogon nardus* Roots, (11) *Cymbopogon nardus* Leaves, (12) *Cymbopogon nardus* Inflorescence, (13) *Sporobolus pyramidalis* Roots, (14) *Sporobolus pyramidalis* Leaves, (15) *Sporobolus africanus* Roots, (16) *Sporobolus africanus* Leaves, (17) *Imperata cylindrical* Roots, (18) *Imperata cylindrical* Leaves, (19) *Cenchrus ciliaris* Roots, (20) *Cenchrus ciliaris* Leaves, (21) *Panicum maximum* Roots, (22) *Panicum maximum* Leaves, (23) *Cymbopogon* spp. Roots, (24) *Cymbopogon* spp. Leaves and (25) *Cynodon dactylon* Whole plant.

**Figure 3 plants-09-00371-f003:**
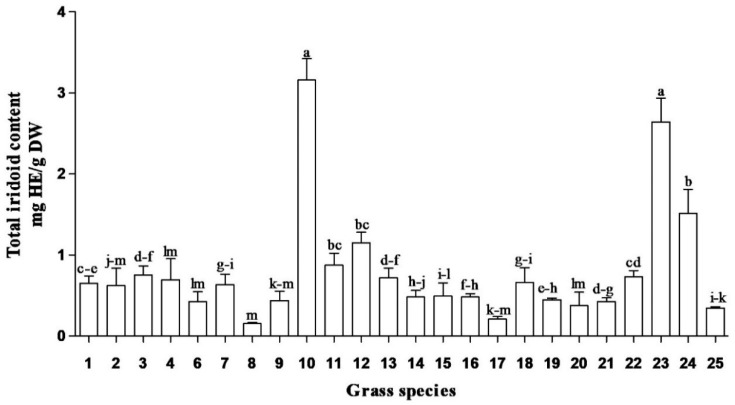
Total iridoid content in grass species used in South African traditional medicine. HE = harpagoside equivalents; DW = dry weight. Bar represents mean ± standard error, *n* = 3. Bars with different letter(s) are significantly (*p* ≤ 0.05) different based on Duncan’s multiple range test (DMRT). (1) *Coix lacryma-jobi* Roots, (2) *Coix lacryma-jobi* Leaves, (3) *Eragrostis curvula* Roots, (4) *Eragrostis curvula* Leaves, (6) *Setaria megaphylla* Roots, (7) *Setaria megaphylla* Leaves, 8. *Vetiveria zizanioides* Roots, 9. *Vetiveria zizanioides* Leaves, (10) *Cymbopogon nardus* Roots, (11) *Cymbopogon nardus* Leaves, (12) *Cymbopogon nardus* Inflorescence, (13) *Sporobolus pyramidalis* Roots, (14) *Sporobolus pyramidalis* Leaves, (15) *Sporobolus africanus* Roots, (16) *Sporobolus africanus* Leaves, (17) *Imperata cylindrical* Roots, (18) *Imperata cylindrical* Leaves, (19) *Cenchrus ciliaris* Roots, (20) *Cenchrus ciliaris* Leaves, (21) *Panicum maximum* Roots, (22) *Panicum maximum* Leaves, (23) *Cymbopogon* spp. Roots, (24) *Cymbopogon* spp. Leaves and (25) *Cynodon dactylon* Whole plant.

**Figure 4 plants-09-00371-f004:**
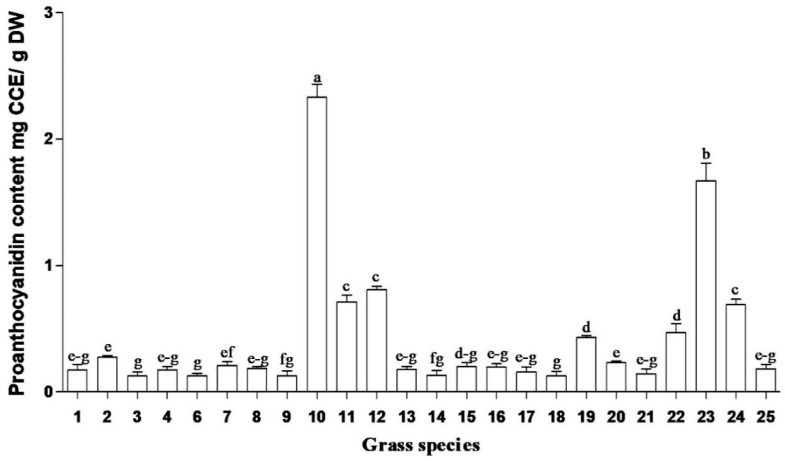
Condensed tannins (Proanthocynidin) in grass species used in South African traditional medicine. CCE = luecocyanidin equivalents; DW = dry weight. Bar represents mean ± standard error, *n* = 3. Bars with different letter(s) are significantly (*p* ≤ 0.05) differences based on Duncan’s multiple range test (DMRT). (1) *Coix lacryma-jobi* Roots, (2) *Coix lacryma-jobi* Leaves, (3) *Eragrostis curvula* Roots, (4) *Eragrostis curvula* Leaves, (6) *Setaria megaphylla* Roots, (7) *Setaria megaphylla* Leaves (8) *Vetiveria zizanioides* Roots, (9) *Vetiveria zizanioides* Leaves, (10) *Cymbopogon nardus* Roots, (11) *Cymbopogon nardus* Leaves, (12) *Cymbopogon nardus* Inflorescence, (13) *Sporobolus pyramidalis* Roots, (14) *Sporobolus pyramidalis* Leaves, (15) *Sporobolus africanus* Roots, (16) *Sporobolus africanus* Leaves, (17) *Imperata cylindrical* Roots, (18) *Imperata cylindrical* Leaves, (19) *Cenchrus ciliaris* Roots, (20) *Cenchrus ciliaris* Leaves, (21) *Panicum maximum* Roots, (22) *Panicum maximum* Leaves, (23) *Cymbopogon* spp. Roots, (24) *Cymbopogon* spp. Leaves and (25) *Cynodon dactylon* Whole plant.

**Table 1 plants-09-00371-t001:** Concentrations (µg/g DW) of hydroxybenzoic acids in 80% methanolic extracts of grasses used in South African traditional medicine. Values represent mean ± standard error, *n* = 3. In each column, values with different letter(s) are significantly (*p* ≤ 0.05) different, based on Duncan’s multiple range test (DMRT). <LOD = below level of detection.

Plant Species	Plant Part		Hydroxybenzoic Acids (µg/g DW)					
		Total Phenolic Content (mg GAE/g DW)	Gallic Acid	*p*-Hydroxy-Benzoic Acid	Protocatechuic Acid	Salicylic Acid	Syringic Acid	Vanillic Acid
*Coix lacryma-jobi*	Roots	9.0 ± 0.95 ^b,c^	<LOD	9.3 ± 1.032 ^d,e^	5.9 ± 0.39 ^b^	1.6 ± 0.14 ^m^	2.9 ± 0.52 ^e,f,g,h,i^	11.6 ± 0.34 ^j,k,l^
	Leaves	12.2 ± 1.38 ^b^	0.4 ± 0.03 ^b,c^	34.2 ± 2.96 ^a^	9.0 ± 0.84 ^a^	1.8 ± 0.13 ^l,m^	3.1 ± 0.23 ^d,e,f,g,h^	34.4 ± 0.24 ^b^
*Cenchrus ciliaris*	Roots	26.6 ± 1.99 ^a^	0.2 ± 0.04 ^d^	10.3 ± 0.21 ^c,d^	0.6 ± 0.08 ^i^	67.4 ± 2.34 ^e^	3.6 ± 0.11 ^c,d^	14.8 ± 0.58 ^h,i,j^
	Leaves	9.0 ± 0.93 ^b,c^	0.5 ± 0.13 ^b^	10.7 ± 0.38 ^c,d^	1.2 ± 0.04 ^f,g,h,i^	248.2 ± 3.875 ^c^	5.4 ± 0.09 ^b^	16.9 ± 0.58 ^f,g,h,i^
*Cynodon dactylon*	Whole plant	6.3 ± 0.32 ^c–f^	0.1 ± 0.00 ^e,f^	3.3 ± 0.18	0.7 ± 0.03 ^h,i^	37.8 ± 1.06 ^g^	1.5 ± 0.08 ^l^	12.6 ± 0.45 ^j,k,l^
*Cymbopogon nardus*	Roots	28.9 ± 1.81 ^a^	0.1 ± 0.00 ^e,f^	9.2 ± 0.18 ^d,e^	3.8 ± 0.06 ^c^	8.2 ± 0.22 ^k,l,m^	2.4 ± 0.10 ^i,j,k^	14.1 ± 0.45 ^i,j,k^
	Leaves	9.1 ± 0.90 ^b–d^	0.1 ± 0.02 ^e,f^	6.9 ± 0.16 ^f,g^	2.5 ± 0.09 ^c,d,e,f,g^	5.3 ± 0.16 ^k,l,m^	5.0 ± 0.24 ^b^	19.6 ± 1.24 ^d,e,f^
	Inflorescence	8.9 ± 0.84 ^b,c^	0.3 ± 0.13 ^c^	11.4 ± 0.49 ^c^	6.4 ± 0.16 ^b^	2.4 ± 0.10 ^l,m^	3.9 ± 0.18 ^c^	18.4 ± 1.49 ^d,e,f,g^
*Cymbopogon* spp.	Roots	30.9 ± 2.93 ^a^	0.1 ± 0.01 ^e,f^	68 ± 0.76 ^f,g^	2.0 ± 0.22 ^d,e,f,g,h,i^	13.2 ± 0.54 ^j,k^	2.9 ± 0.33 ^e,f,g,h,i^	11.1 ± 0.93 ^k,l^
	Leaves	9.8 ± 0.38 ^b,c^	0.8 ± 0.21 ^a^	6.7 ± 0.15 ^f,g^	2.9 ± 0.13 ^c,d,e^	4.6 ± 0.07 ^k,l,m^	2.7 ± 0.20 ^f,g,h,i,j^	18.5 ± 0.51 ^d,e,f,g^
*Eragrostis curvula*	Roots	5.3 ± 0.13 ^d–f^	0.1 ± 0.02 ^e,f^	9.8 ± 0.34 ^c,d^	1.8 ± 0.09 ^d,e,f,g,h,i^	26.7 ± 0.42 ^h,i^	3.3 ± 0.37 ^d,e,f,g^	46.05 ± 2.38 ^a^
	Leaves	7.4 ± 0.53 ^b–d^	0.2 ± 0.03 ^d^	5.3 ± 0.55 ^g^	2.1 ± 0.13 ^d,e,f,g,h^	30.5 ± 1.33 ^g,h^	3.3 ± 0.40 ^c,d,e,f,g^	21.7 ± 1.3 ^c,d^
	Inflorescence	-	0.1 ± 0.00 ^e,f^	10.7 ± 0.79 ^c,d^	9.0 ± 0.38 ^a^	7.3 ± 0.18 ^k,l,m^	7.0 ± 0.53 ^a^	36.0 ± 1.12 ^b^
*Imperata cylindrica*	Roots	6.4 ± 0.49 ^c–f^	0.3 ± 0.06 ^c^	6.9 ± 0.19 ^f,g^	3.1 ± 0.13 ^c,d^	12.1 ± 0.49 ^j,k,l^	3.4 ± 0.27 ^c,d,e^	24.4 ± 1.24 ^c^
	Leaves	7.3 ± 0.14 ^b–d^	0.2 ± 0.01 ^d,e^	6.834 ± 0.436 ^f,g^	3.1 ± 0.07 ^c,d^	19.3 ± 0.77 ^i,j^	2.7 ± 0.12 ^g,h,i,j^	24.0 ± 1.45 ^c^
*Panicum maximum*	Roots	6.9 ± 0.24 ^c,d^	0.1 ± 0.01 ^e,f^	9.9 ± 0.43 ^c,d^	2.9 ± 0.10 ^c,d,e^	47.6 ± 2.63 ^f^	3.2 ± 0.09 ^d,e,f,g,h^	21.4 ± 0.95 ^c,d,e^
	Leaves	9.8 ± 1.00 ^b,c^	0.3 ± 0.02 ^c,d^	10.26 ± 0.197 ^c,d^	6.9 ± 2.70 ^b^	356.82 ± 12.92 ^b^	3.1 ± 0.19 ^d,e,f,g,h^	14.6 ± 0.41 ^h,i,j,k^
*Setaria megaphylla*	Roots	8.0 ± 0.49 ^b–d^	0.1 ± 0.01 ^e,f^	11.2 ± 1.32 ^c^	1.6 ± 0.29 ^e,f,g,h,i^	93.1 ± 6.12 ^d^	3.4 ± 0.56 ^c,d,e^	23.5 ± 4.77 ^c^
	Leaves	8.6 ± 0.62 ^b–d^	0.2 ± 0.021 ^d,e^	14.0 ± 1.55 ^b^	2.5 ± 0.26 ^c,d,e,f^	467.3 ± 15.36 ^a^	3.3 ± 0.35 ^c,d,e,f^	21.1 ± 3.67 ^c,d,e^
*Sporobolus africanus*	Roots	4.2 ± 0.21 ^f^	0.1 ± 0.01 ^e,f^	6.0 ± 0.29 ^f,g^	1.9 ± 0.04 ^d,e,f,g,h,i^	2.9 ± 0.05 ^k,l,m^	2.2 ± 0.17 ^j,k^	10.4 ± 0.41 ^l^
	Leaves	6.7 ± 0.45 ^c–f^	0.1 ± 0.05 ^e,f^	6.0 ± 0.15 ^f,g^	1.1 ± 0.08 ^g,h,i^	5.2 ± 0.10 ^k,l,m^	2.0 ± 0.15 ^k,l^	12.5 ± 0.29 ^j,k,l^
*Sporobolus pyramidalis*	Roots	4.4 ± 0.45 ^e,f^	<LOD	6.2 ± 0.22 ^f,g^	1.2 ± 0.13 ^f,g,h,i^	5.3 ± 0.19 ^k,l,m^	3.4 ± 0.28 ^c,d,e^	20.8 ± 1.84 ^c,d,e^
	Leaves	6.4 ± 0.29 ^c–f^	0.2 ± 0.06 ^d,e^	7.7 ± 0.23 ^e,f^	1.5 ± 0.10 ^f,g,h,i^	4.7 ± 0.04 ^k,l,m^	2.5 ± 0.06 ^h,i,j,k^	16.7 ± 1.24 ^f,g,h,i^
*Vetiveria zizanioides*	Roots	7.3 ± 0.28 ^b–d^	0.1 ± 0.01 ^e,f^	3.5 ± 0.15 ^h^	1.4 ± 0.01 ^f,g,h,i^	2.6 ± 0.06 ^l,m^	3.3 ± 0.20 ^c,d,e,f^	15.1 ± 0.86 ^g,h,i,j^
	Leaves	6.0 ± 0.46 ^c–f^	0.1 ± 0.010 ^e,f^	7.2 ± 0.18 ^f,g^	1.8 ± 0.09 ^d,e,f,g,h,i^	4.6 ± 0.13 ^k,l,m^	3.6 ± 0.14 ^c,d^	17.8 ± 0.58 ^e,f,g,h^

- Not evaluated.

**Table 2 plants-09-00371-t002:** Quantity (µg/g DW) of hydroxycinnamic acids in 80% methanolic extracts of different grasses. Values represent mean ± standard error, *n* = 3. In each column, values with different letter(s) are significantly (*p* ≤ 0.05) different, based on Duncan’s multiple range test (DMRT). <LOD = below level of detection.

Plant Species	Plant Part		Hydroxycinnamic Acids (µg/g DW)				
		Total Phenolic Content (mg GAE/g DW)	Caffeic Acid	Chlorogenic Acid	*p*-Coumaric Acid	Ferulic Acid	Sinapic Acid
*Coix lacryma-jobi*	Roots	9.0 ± 0.95 ^b,c^	14.1 ± 1.39 ^c^	144.8 ± 11.28 ^a^	59.8 ± 7.31 ^b^	22.9 ± 5.28 ^d^	<LOD
	Leaves	12.2 ± 1.38 ^b^	16.3 ± 1.50 ^b^	105.3 ± 8.50 ^b^	80.4 ± 5.45 ^a^	26.8 ± 2.31 ^c^	0.3 ± 0.03 ^m^
*Cenchrus ciliaris*	Roots	26.6 ± 1.99 ^a^	0.3 ± 0.03 ^i,j^	<LOD	34.6 ± 2.44 ^e,f^	9.4 ± 0.57 ^h,i,j^	0.9 ± 0.04 ^f,g^
	Leaves	9.0 ± 0.93 ^b,c^	0.4 ± 0.04 ^i,j^	<LOD	32.2 ± 0.51 ^e,f^	9.0 ± 0.21 ^h,i,j^	2.4 ± 0.07 ^a^
*Cynodon dactylon*	Whole plant	6.3 ± 0.32 ^c–f^	0.1 ± 0.02 ^j^	0.2 ± 0.02 ^f^	23.1 ± 0.74 ^h,i^	7.5 ± 0.23 ^j,k^	0.6 ± 0.03 ^i,j,k^
*Cymbopogon nardus*	Roots	28.9 ± 1.81 ^a^	18.0 ± 0.24 ^a^	35.5 ± 1.20 ^f^	64.0 ± 0.33 ^b^	56.0 ± 0.74 ^a^	2.1 ± 0.015 ^c^
	Leaves	9.1 ± 0.90 ^b–d^	3.6 ± 0.64 ^f^	4.6 ± 0.64 ^f^	51.4 ± 0.66 ^c^	14.3 ± 1.48 ^f,g^	0.6 ± 0.03 ^i,j,k^
	Inflorescence	8.9 ± 0.84 ^b,c^	1.1 ± 0.07 ^h,i,j^	0.1 ± 0.01 ^f^	24.2 ± 0.85 ^g,h,i^	15.3 ± 0.37 ^f^	0.9 ± 0.03 ^f^
*Cymbopogon* spp.	Roots	30.9 ± 2.93 ^a^	10.8 ± 1.11 ^d^	12.3 ± 1.70 ^d,e^	60.3 ± 3.45 ^b^	33.1 ± 1.98 ^b^	1.6 ± 0.18 ^d^
	Leaves	9.8 ± 0.38 ^b–e^	1.4 ± 0.05 ^g,h,i^	3.6 ± 0.71 ^f^	25.5 ± 0.73 ^g,h^	8.0 ± 0.22 ^i,j,k^	0.4 ± 0.02 ^l,m^
*Eragrostis curvula*	Roots	5.3 ± 0.13 ^d–f^	1.5 ± 0.132 ^g,h,i^	0.3 ± 0.01 ^f^	19.2 ± 0.69 ^i,j^	24.4 ± 0.90 ^c,d^	1.2 ± 0.14 ^e^
	Leaves	7.4 ± 0.53 ^b–d^	1.3 ± 0.11 ^g,h,i,j^	3.3 ± 0.26 ^f^	22.0 ± 1.69 ^h,i,j^	17.1 ± 1.40 ^e,f^	1.4 ± 0.18 ^d^
	Inflorescence	-	4.1 ± 0.44 ^f^	0.7 ± 0.02 ^f^	16.6 ± 0.40 ^j,k^	21.7 ± 1.58 ^d^	0.8 ± 0.08 ^f,g,h^
*Imperata cylindrica*	Roots	6.4 ± 0.49 ^c–f^	1.5 ± 0.12 ^g,h,i^	15.3 ± 1.00 ^d^	32.0 ± 1.89 ^e,f^	9.7 ± 0.43 ^h,i,j^	1.0 ± 0.09 ^e,f^
	Leaves	7.3 ± 0.14 ^b–d^	1.9 ± 0.15 ^g,h^	14.2 ± 0.58 ^d^	29.2 ± 1.01 ^f,g^	8.8 ± 0.51 ^h,i,j,k^	0.5 ± 0.01 ^j,k,l,m^
*Panicum maximum*	Roots	6.9 ± 0.24 ^c,d^	0.7 ± 0.01 ^i,j^	1.8 ± 0.17 ^f^	46.3 ± 1.19 ^c,d^	10.6 ± 0.33 ^h,i,j^	0.7 ± 0.03 ^g,h,i^
	Leaves	9.8 ± 1.00 ^b,c^	2.3 ± 0.09 ^g^	3.0 ± 0.15 ^f^	35.5 ± 1.94 ^e^	10.4 ± 0.14 ^h,i,j^	0.5 ± 0.01 ^k,l,m^
*Setaria megaphylla*	Roots	8.0 ± 0.49 ^b–d^	0.5 ± 0.07 ^i,j^	1.9 ± 0.37 ^f^	44.7 ± 4.14 ^d^	9.7 ± 1.18 ^h,i,j^	0.6 ± 0.08 ^i,j,k^
	Leaves	8.6 ± 0.62 ^b–d^	2.5 ± 0.46 ^g^	6.9 ± 0.48 ^e,f^	42.8 ± 4.70 ^d^	15.5 ± 1.45 ^f^	0.6 ± 0.09 ^i,j,k^
*Sporobolus africanus*	Roots	4.2 ± 0.21 ^f^	6.4 ± 0.48 ^e^	0.1 ± 0.01 ^f^	10.4 ± 0.88 ^l^	5.7 ± 0.30 ^k^	0.3 ± 0.05 ^l,m^
	Leaves	6.7 ± 0.45 ^c–f^	0.6 ± 0.03 ^i,j^	0.1 ± 0.00 ^f^	20.1 ± 0.76 ^h,i,j^	11.7 ± 0.21 ^g,h^	0.7 ± 0.03 ^h,i,j^
*Sporobolus pyramidalis*	Roots	4.4 ± 0.45 ^e,f^	0.5 ± 0.07 ^i,j^	< LOD	11.0 ± 0.60 ^l^	18.7 ± 0.30 ^e^	0.4 ± 0.11 ^l,m^
	Leaves	6.4 ± 0.29 ^c–f^	0.6 ± 0.05 ^i,j^	< LOD	12.2 ± 0.41 ^k,l^	11.2 ± 0.44 ^h,i^	0.5 ± 0.06 ^i,j,k,l^
*Vetiveria zizanioides*	Roots	7.3 ± 0.28 ^b–d^	1.4 ± 0.04 ^g,h,i^	1.6 ± 0.29 ^f^	30.8 ± 0.33 ^e,f^	22.1 ± 0.71 ^d^	2.2 ± 0.177 ^b^
	Leaves	6.0 ± 0.46 ^c–f^	0.8 ± 0.05 ^h,i,j^	2.1 ± 0.13 ^f^	46.3 ± 0.50 ^c,d^	10.7 ± 0.25 ^h,i,j^	0.4 ± 0.05 ^l,m^

- Not evaluated.

**Table 3 plants-09-00371-t003:** DPPH (1,1-Diphenyl-2-picrylhydrazyl) radical-scavenging activity (EC_50_) of grasses evaluated for their therapeutic potential in South African traditional medicine. Values indicate mean ± SE, *n* = 3. Different letter(s) associated with EC_50_ indicate significant differences at the 5% level of significance. Ascorbic acid (positive control) = 0.025 ± 0.004.

Grass Species	Plant Part	EC_50_ (mg/mL)
*Coix lacryma-jobi*	Roots	0.09 ± 0.017 ^a,b,c,d,e^
	Leaves	0.10 ± 0.004 ^a,b,c^
*Cenchrus ciliaris*	Roots	0.09 ± 0.006 ^a,b,c,d,e^
	Leaves	0.09 ± 0.011 ^a,b,c,d,e^
*Cynodon dactylon*	Whole plant	0.10 ± 0.005 ^a,b,c,d^
*Cymbopogon nardus*	Roots	0.02 ± 0.007 ^h^
	Leaves	0.06 ± 0.002 ^f,g^
	Inflorescence	0.04 ± 0.012 ^f,g^
*Cymbopogon* spp.	Roots	0.04 ± 0.026 ^g,h^
	Leaves	0.08 ± 0.017 ^d,e,f^
*Eragrostis curvula*	Roots	0.11 ± 0.012 ^a,b^
	Leaves	0.08± 0.019 ^c,d,e,f^
*Imperata cylindrica*	Roots	0.10 ± 0.000 ^a,b,c,d,e^
	Leaves	0.09 ± 0.001 ^a,b,c,d,e^
*Panicum maximum*	Roots	0.11 ± 0.021 ^a,b,c^
	Leaves	0.08 ± 0.011 ^b,d,e,f^
*Setaria megaphylla*	Roots	0.11 ± 0.017 ^a^
	Leaves	0.07 ± 0.005 ^e,f^
*Sporobolus africanus*	Roots	0.10 ± 0.010 ^a,b,c,d^
	Leaves	0.10 ± 0.002 ^a,b,c,d^
*Sporobolus pyramidalis*	Roots	0.08 ± 0.014 ^b,d,e,f^
	Leaves	0.08 ± 0.001 ^b,d,e,f^
*Vetiveria zizanioides*	Roots	0.11 ± 0.004 ^a,b,c^
	Leaves	0.06 ± 0.014 ^f,g^

**Table 4 plants-09-00371-t004:** Ferric-reducing antioxidant power (FRAP) of grasses used in South African traditional medicine.

Grass Species	Plant Part	Slope	R^2^
*Coix lacryma-jobi*	Roots	0.054 ± 0.003	0.6896
	Leaves	0.060 ± 0.002	0.6723
*Cenchrus ciliaris*	Roots	0.071 ± 0.002	0.7935
	Leaves	0.069 ± 0.0032	0.7817
*Cynodon dactylon*	whole plant	0.078 ± 0.001	0.8988
*Cymbopogon nardus*	Roots	0.070 ± 0.005	0.5142
	Leaves	0.079 ± 0.014	0.6405
	Inflorescence	0.074 ± 0.002	0.6545
*Cymbopogon* spp.	Roots	0.069 ± 0.003	0.6317
	Leaves	0.082 ± 0.007	0.7303
*Eragrostis curvula*	Roots	0.070 ± 0.004	0.7879
	Leaves	0.051 ± 0.005	0.6267
*Imperata cylindrica*	Roots	0.079 ± 0.005	0.7213
	Leaves	0.062 ± 0.009	0.6726
*Panicum maximum*	Roots	0.076 ± 0.003	0.8495
	Leaves	0.077 ± 0.001	0.7304
*Setaria megaphylla*	Roots	0.073 ± 0.002	0.7927
	Leaves	0.059 ± 0.001	0.6646
*Sporobolus africanus*	Roots	0.124 ± 0.004	0.8667
	Leaves	0.086 ± 0.001	0.8547
*Sporobolus pyramidalis*	Roots	0.094 ± 0.003	0.8509
	Leaves	0.081 ± 0.003	0.6991
*Vetiveria zizanioides*	Roots	0.071 ± 0.007	0.825
	Leaves	0.069 ± 0.012	0.7591
Ascorbic acid (positive control)		0.095 ± 0.001	0.999

Values are represented as mean ± standard error (*n* = 3). A higher slope value corresponds to better ferric-reducing power of the sample. The measure of reliability, using r^2^, indicates the fitness of the curve, and the closer the value of the sample to 1, the higher the reliability.

**Table 5 plants-09-00371-t005:** The sequence of linear gradients and isocratic flows of solvents in the mobile phase of the reversed-phase UHPLC.

Sequence	Solvent	Duration (min)
1	5% B	0.8
2	5%–10% B	0.4
3	Isocratic 10% B	0.7
4	10%–15% B	0.5
5	Isocratic 15% B	1.3
6	15%–21% B	0.3
7	Isocratic 21% B	1.2
8	21%–27% B	0.5
9	27%–50% B	2.3
10	50%–100% B	1.0
11	100%–5%	0.5

Solvent B: Acetonitrile, balanced with 7.5 mM formic acid, at a flow rate of 250 μL·min^−1^.

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
