# Peer review of "Phytochemical Profiles and Antioxidant Activity of Grasses Used in South African Traditional Medicine"

_plants, 2020, doi:10.3390/plants9030371_

Round 1

Reviewer 1 Report

This submission reports the results of the phytochemical study and DPPH-scavenging activity of 12 weed plants  (Poaceae family) from South Africa is presented. After detailed reading, I found manuscript prepared in the consisted manner and all parts: abstract, introduction, methodology, results and discussion are presented appropriately. References are appropriate to present paper and correctly cited. The conclusion is based on obtained results and supports the assumed hypothesis. Undoubtedly this manuscript contributes to the better knowledge of Poaceae plants in South Africa and opens the possibility of their uses as the source of bioactive molecules. Although this article is of moderate originality and scientific level, it has considerable applicative potential.

Summarized I found this manuscript written in the proper manner but with average scientific value and novelty.

However, there are quite a few points that need to be addressed to make the manuscript more acceptable.  

Introduction:

The reason why particular grass species were chosen for the current study, should be inserted in the introduction part, although it is obvious that this study is based on the two review papers, previously published by the same author  (ref, 13 and 14).

Besides,  in the introduction part, the novelty of this research should be highlighted.

Results and discussion:

I have not found results of antioxidant activity for investigated plant species in reference 21. (p3, line 101).

Concerning applied assays,  one should take into account that F-C assay for TPC is not very specific since numerous compounds (reducents) are shown to give positive results, incl. ascorbic acid and other vitamins, amino acids and peptides, guanine, some sugars... The F-C assay can, thus, only be used as a measure of reducing power.

Furthermore,  it should be highlighted that both Cymbopogon sp., are distinguished by the highest content of most evaluated classes of biomolecules. (p. 3, line 108).

Concerning the content of evaluated phenolic acid, it will be clearer if you add one column with total phenolic acids content. (table 1 and 2).

I suggest to authors to try to distinguish evaluated species regarding the content of carboxylic acids from one side and cinnamic acids ( phenylpropanoids derivatives) from the others. Cluster analysis or PCA (principle compounds analysis) can be useful)

Information on antioxidant activity can not be verified by only one simple DPPH test. 

Material and Methods:

You have selected two samples of Cymbopogon genus: Cymbopogon nardus and Cymbopogon spp. what exactly are the later ones? Please describe it.

What was the ratio plant material/80% MeOH for preparing extracts for HPLC?   

What was the principle of the method for flavonoid determination?

The method for HPLC/MS should be given in more details, at least as a very well known DPPH test. Did you detect some more phenolic acids besides those in the standard mixture?

Summarized I found this manuscript written in the proper manner, with a lot of results but with average scientific value and novelty

Author Response

Introduction

The reason why particular grass species were chosen for the current study, should be inserted in the introduction part, although it is obvious that this study is based on the two review papers, previously published by the same author (ref, 13 and 14).

Reason why particular grasses were chosen?

The grasses were mentioned to be prominent in traditional medicines during an ethnobotanical survey as we have previously shown. The reasons for selecting these species has now been included in the introduction, lines 65-68

Besides, in the introduction part, the novelty of this research should be highlighted.

It was highlighted that not a lot of work has been done on grasses as remedies, thus little scientific evidence is available for their use in traditional medicine. Lines 68-72

I have not found results of antioxidant activity for the investigated plant species in reference 21. (P3, line 101)

The antioxidant activity is presented in Table 1 as the inhibition (% In) of Methyl Linoleate Oxidation.

The antioxidant activity was expressed as percentage (%) inhibition of the formation of MeLo-conjugated diene hydroperoxides after 72 h of oxidation

Results and discussion:

Concerning applied assays, one should take into account that F-C assay for TPC is not very specific since numerous compounds (reducents) are shown to give positive results, incl. ascorbic acid and other vitamins, amino acids and peptides, guanine, some sugars... The F-C assay can, thus, only be used as a measure of reducing power.

Furthermore, it should be highlighted that both Cymbopogon sp., are distinguished by the highest content of most evaluated classes of biomolecules. (p. 3, line 108). It was further emphasised in lines 82-84

Concerning the content of evaluated phenolic acid, it will be clearer if you add one column with total phenolic acids content. (table 1 and 2). The column was added in table 1 and 2

I suggest to authors to try to distinguish evaluated species regarding the content of carboxylic acids from one side and cinnamic acids (phenylpropanoids derivatives) from the others. Cluster analysis or PCA (principle compounds analysis) can be useful)

We are grateful to the reviewer for the effort made in improving our manuscript, however we had challenges with the suggested analyses (Cluster and PCA analysis).

We attempted to analyse the data with both approaches suggested, although the PCA does show that there is variation, this was only observed for two of the components which accounted for 51% of the variation. As such, there was no clear pattern that we observed from these analyses. We have attached this analyses as a supplementary file for the reviewer’s information.

Additionally, based on the current dataset we were unable to compare carboxylic acids and cinnamic acids to distinguish between the species. If the reviewer could provide more clarity on this analyses we will be happy to perform it.

Information on antioxidant activity cannot be verified by only one simple DPPH test. 

Additional antioxidant activity results were incorporated (Ferric reducing power). Line 257-277

Material and Methods:

You have selected two samples of Cymbopogon genus: Cymbopogon nardus and Cymbopogon spp. what exactly are the later ones? The latter is an unidentified species of Cymbopogon, the reason why it was not identified was because it was not flowering during the time of collection. It is very difficult to confirm the species of a glass plant without the inflorescence.

Please describe it.

The grass plant had a different height as compared with other the Cymbopogon species collected in this study, the leaves smelled different and it produced a different colour of extract.

What was the ratio plant material/80% MeOH for preparing extracts for HPLC? 13.3 mg/ml (20 mg/1.5 ml) line 341

What was the principle of the method for flavonoid determination? Spectrophotometric assay based on aluminium complex formation (lines 327-328)

The method for HPLC/MS should be given in more details, at least as a very well known DPPH test. Did you detect some more phenolic acids besides those in the standard mixture?

The UHPLC/MS method was amended and was described with more detail (lines 362-378)

Summarized I found this manuscript written in the proper manner, with a lot of results but with average scientific value and novelty

Reviewer 2 Report

The manuscript entitled “Phytochemical profiles and antioxidant activity of grasses used in South African traditional medicine” of the authors Gebashe , Aremu , Gruz , Finnie , and Van Staden  deals with the interesting topic of assessing the phytochemical composition and antioxidant activity of plants used in traditional medicine and/or as nourishing food. In particular, they evaluate the content of phenolics, flavonoids, proanthocyanidins, iridoids, and antioxidant activity, as well as the profiles of phenolic acids in the different organs of 12 species of grasses used in South African traditional medicine.

Although the originality is not high, the research seems well executed, the results are sound and reliable, and could be of interest for the scientific community working on this topic. The conclusions are fully justified by the data. Nevertheless, in my opinion, the manuscript require some revisions prior to be accepted for publication in Plants. The major weakness of the work is the fact that in grasses most phenols are bound to the polymeric constituents of the cell wall, so that their amounts could have been strongly underestimated. Usually an evaluation of soluble, soluble-bound and insoluble phenols is performed. This would make the manuscript more appealing and complete. Anyway, I suggest, at least, to change "total phenolic" with "total soluble phenolic" all along the text. I also suggest giving an explanation about the reasons why insoluble phenols have not been assessed. These give an important contribute to the healthy properties of grasses and other plants. This should be clearly stated in the text and supported by references.

In few points, the manuscript is unclear and require improvement.

See the attached file for detailed corrections and specific points to be addressed.

Author Response

Line by Line response to Comments

Introduction:

Line 47 Herbivory changed to herbivores

Line 66-67 ‘A sizeable number (more than 100) of…..’ was changed to be more direct “More than 100 secondary metabolites……”

Line 67 ‘Some of’ was deleted to start the sentence with “These..”

Line 69 The sentence was connected with an ‘and’ to become one sentence.

Results and Discussion

Line 73 In grasses, most phenols are usually bound to the polymeric constituents of the cell wall. For this reason, I suggest to change "total phenolic" with "total soluble phenolic", here and elsewhere in the manuscript. I also suggest giving an explanation about the reasons why bound phenols have not been assessed. These give an important contribute to the healthy properties of grasses and other plants. At this regard please see the following papers:

Mastrangelo et al., 2009. Planta, 229: 343. https://doi.org/10.1007/s00425-008-0834-x

Laddomada et al., 2017. Genet Resour Crop Evol, 64: 587. https://doi.org/10.1007/s10722-016-0386-z

-“Total phenolic content’ was changed to ‘total soluble phenolic content’ throughout the manuscript see, e.g. lines 23, 75, 117, 303, 319-323

The authors are grateful for the citations as they were informative and the explanation of only evaluating soluble phenolics was provided in the introduction see lines 68-72. We understand the point of the reviewer and we are grateful for the insight, thus for future studies bound phenols will be evaluated in our studies.

The bound phenolic acids have been considered less important as compared to free forms and also the fact that are scarcely digested in the stomach or intestine contributes to them not been given attention. However, it has been discovered that by reaching the intestines intact they act as potent antioxidants and anti-inflammatory mediators, also reduce risk of colorectal cancer (Laddomada et al 2016).

At their first occurrence, please add the Authority to the binomial name of each species. Furthermore, in my opinion a figure of the plants/organs assayed would make the manuscript more appealing to the reader.

-Authority of the binomial names was inserted for the different species in lines 77, 87, 89,104-105 and 110-111.

- Some of the grass species assayed are attached as an appendix in this MS

The meaning is unclear.

Line 103 was rephrased for better clarity see line 109-110.

I suggest to make a correlation analysis between each class of compounds and the antioxidant activity in order to see which one of the compounds contributes most to the antioxidant power.

We are grateful for the comments and suggestions on the improvement of our MS. However, under these circumstances, a correlation investigation is not feasible. Since the phenolic acids were not evaluated individually and their quantity differs within the extracts, it would be difficult to determine which phenolic acid was responsible for the said activity. It may be that a compound with a low content was more active than a compound that is more abundant and this will result in a bias. As such we have instead conducted a correlation to determine if the total phenolics content did influence antioxidant activity.

Correlation was done using spss 26 and results incorporated in the results section, see line 207-209

Materials and Methods

Please define the number of replicas.

Replicates inserted in the MS (line 300)

This section should be moved at the beginning of the "Materials and Methods" section and ferred to all applied methods. Thus it should include all solvents, chemicals and standards.

The section was moved and amended as suggested (Lines 280-290)

It is not clear how these different concentrations were prepared. I can assume by weighting the dried extracts and resuspending them in 80% methanol. Please explain the procedure in the text.

- The methods were rephrased to be more clear (lines 385-386)

Conclusion

Unclear, please rephrase.

Sentence was rephrased (lines 438-439)

Round 2

Reviewer 1 Report

After reading the latest version of the manuscript and considering authors responses and explanations, I found that authors accepted most of my comments and corrected manuscript in regards to. Therefore I suggest to the Editor to accept the revised version of this manuscript.